# Health and well-being after being deployed in a major incident; how do Swedish ambulance nurses perceive their health recover process? A qualitative study

Karin Blomberg, Karin Hugelius 

Faculty of Medicine and Health, Örebro University, Orebro, Sweden

**Correspondence to**
Karin Hugelius;
Karin.hugelius@oru.se

## ABSTRACT

**Objectives** To explore health problems and the recovery process after being deployed in a major incident.

**Design** Qualitative, explorative design.

**Setting** Ambulance services in Sweden.

**Participants and methods** Semistructured, individual two-session interviews with 15 ambulance nurses with the experience of being deployed to major incidents were conducted. Data were analysed with thematic analysis.

**Results** Being deployed in major incidents was perceived to be straining and led to both physical health problems and distress. To recover, the ambulance nurses strived to use strategies to distance themselves from the situation and created supportive conditions for their recovery, and if successful, the experiences led to both professional and personal growth and self-awareness. However, being deployed in major incidents without significant preparedness or experience could harm individuals and, in the worst case, end their career.

**Conclusions** A successful recovery from the physical and mental exhaustion experienced after being deployed in a major incident required both individual abilities and self-care strategies as well as a supportive working environment. Supporting individual recovery strategies and following up on physical and mental well-being over time should be part of all ambulance services procedures after major incidents.

## STRENGTHS AND LIMITATIONS OF THIS STUDY

⇒ The two-session interviews used stimulated a deeper reflection and rich material to analyse.

⇒ Information power was considered as sufficient.

⇒ Data were analysed and confirmed by researchers with different perspectives on the topic.

⇒ It can never be excluded that another study sample, the characteristics of the major events forming the basis of this study or the timing of the data collection might have influenced the results.

## BACKGROUND

Ambulance personnel are, daily, subject to situations that can evoke distress, but also occasional events such as major incidents, mass casualty situations or disasters that might adversely affect their well-being.[1 2] Responding to major incidents, such as large accidents, terrorist attacks or natural disasters, is a challenging experience even for experienced and trained professionals. Previous studies suggest that such experiences can have both physical and psychological health consequences among first responders.[3–5] After the 9/11 terrorist attack, musculoskeletal problems, breathing problems and undefined bodily pain were reported among the first responders.[5] Among mental health problems, post-traumatic stress disorder (PTSD) seems to be the most frequently studied condition, and the risk for PTSD has been found to be higher among ambulance personnel than in the general population.[3 6] In particular, professionals in a command position during a major incident or disaster have been found to have an increased risk for mental health problems.[3]

Resilience has been described as the ability to 'bounce back' after a displacement.[7] It does not necessarily mean the absence of stress reactions but that the person recovers and can sometimes grow from these experiences.[8] Resilience should be seen as a process that is highly dependent on the setting around the person, such as social network, work organisational perspectives and cultural and community aspects.[8] Being resilient does not necessarily mean the absence of health problems or pathology but being able to cope with the situation and actively adapt in a healthy way.[9] The resilience of ambulance nurses after being deployed in major incidents is concerning for the individual affected, from an employer perspective and for the broader community that relies on their services.

Especially in the aftermath of a major incident, the medical resources might be stretched, and therefore, it is of extra importance that ambulance nurses are resilient and stay healthy after such events. However, the knowledge on how to promote resilience among ambulance personnel after being deployed in major incidents is still limited. Therefore, this study aimed to explore ambulance nurses' perceived health and the recovery process after being deployed in major incidents.

## METHODS

### Design
An explorative, qualitative study was conducted.

### Study setting
The study was conducted in Sweden. In Sweden, all emergency ambulances are staffed with at least one registered nurse, most often specialised in prehospital emergency care (called ambulance nurse) and one paramedic or a second nurse. In this study, a major incident was defined as an incident that forced the healthcare system to heighten the level of response due to the event.

### Study sample and recruiting
The data used in this study were gathered simultaneously as data for another study, using the same study sample.[10] Nurses or specialised nurses within the ambulance services were recruited by invitations via social media, in official or private groups specifically focused on Swedish ambulance personnel (national wide). In addition, invitations were distributed through ambulance managers in healthcare regions where a major incident had occurred recently. Inclusion criteria were having at least 1 year of working experience as a nurse within the ambulance services and having first-hand experiences from being deployed in a major incident during the last 5 years. Exclusion criteria were personal or professional relationships with any of the authors and being a paramedic (not a nurse). Eligible participants who wanted to participate in the study were asked to read full study information posted on a web page and to fill in a form with their contact information. All nurses who showed interest were contacted by a researcher (KH) to set up a time for the interview. A total of 15 nurses responded to the invitation and were included in the study. No one who volunteered to participate was excluded.

### Data collection
Two-session individual interviews,[11] (n=27) were conducted face to face (n=2) or by phone (n=25) (depending on geographical distance and the choice of the study person) with 15 nurses. All the interviews were conducted by one researcher (KH) during the period of December 2018 to February 2020 using a semistructured interview guide. The interview guide covered questions on experiences working in a major incident, preparations for such experiences, perceived health

effects and recovery strategies after such experiences. The interview guide was tested in two pilot interviews, after which only minor corrections were made. The pilot interviews were therefore included in the analysis. The study participants were asked to focus on one specific major incident, even though many of the participants expressed experiences gained from several major incidents. Probing questions, for example, 'What do you mean by that…' or 'Can you further describe…', were used to obtain a deeper understanding of their experiences. The second interview was conducted within 14 days of the first interview and aimed to follow-up on the dialogue but also to follow-up on the participant's well-being. Three participants declined a second interview due to personal circumstances such as a planned vacation, which resulted in a data set of 27 interviews in total. All interviews were audiorecorded and transcribed verbatim by a professional transcriber. The COREQ checklist was used to report the study.

### Analysis
An inductive thematic analysis was used to analyse the data.[12] After transcription, the whole texts were read through several times to get familiar with the data, as the researcher (KB) primarily responsible for the analysis was not involved in data collection. Thereafter, meaning units responding to the aim of the study were extracted and coded by the first author (KH). The codes were thereafter clustered into subthemes and themes reflecting both the coded extracts and the entire data set. This clustering formed a broader understanding of patterns within the phenomenon studied. During this phase, two researchers participated (KB and KH). A final naming of the themes and subthemes was determined during discussions between the two researchers. Finally, the results were formulated, and quotations (from n=7 participants) were selected to illustrate the findings and to increase trustworthiness.

### Ethical considerations
All participants had received written and verbal study information that their participation was voluntary and that they had the right at any time to refrain from further participation without specifying any reasons. Before the first interview, informed consent was obtained. In all, this paper presents quotations from seven study participants. Because major incidents are rare in Sweden, no information on the study participants or the exact situation behind the specific quotation was provided to protect the participants' identity. To detect signs of potential traumatisation or need for professional counselling, all study participants were asked to respond to the Post-traumatic Checklist-Civilian (PCL-IV), Swedish version. A cut-off value of over 40 was an indicator for remaining stress.[13]

### Patient and public involvement
No patients were involved in this study.

**Table 1** Overview of the findings

| Main themes | Being exhausted and vulnerable | Getting distance from the situation | Establishing supportive conditions for recovery | Personal and professional development |
|---|---|---|---|---|
| Subthemes | Experiencing physical health problems | Using self-reflection | Getting the whole picture | Professional growth |
| | Experiencing mental health problems | Transforming from professional to personal | Creating social support | Personal growth |
| | Feeling vulnerable | | Experiencing professional support | Failure to cope and recover |

## RESULTS

All nurses who volunteered to participate in the study were included. In all, 15 ambulance nurses, participated in the 27 interviews. The interviews lasted between 2 and 89 min (first interviews 24–89 min, median 52 min; second interviews 2–22 min, median 5 min). Of the study participants, 10 were males and five females, aged 29–62 years old (median age 41), with professional experience in ambulance services varying from 1 to 39 years (median 18 years). The major incidents covered in the study included traffic accidents, fires, terrorist attacks and explosions. None of the participants screened indicated remaining traumatic stress according to the PCL-IV. Table 1 provides an overview of the main theme and subthemes.

The experiences of health problems and recovery after being deployed in a major incident could be described as a process starting right after the deployment with being physically and mentally exhausted and vulnerable, continuing with a phase of striving to find strategies that provided distance from the situation and supportive conditions for recovery, followed by an incorporation of the experiences into both personal and professional development (see figure 1).

### Being exhausted and vulnerable

After finalising the deployment in the major incident, all nurses described feelings of being totally exhausted, both physically and mentally. Physical health problems such as a general bodily pain, headache or dizziness were most intense during the first few days. A tiredness that was

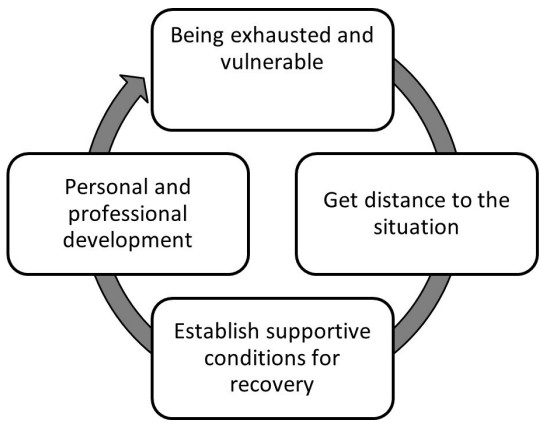

**Figure 1** The process of recovery after deployment in a major incident.

overwhelming was described; at the same time, sleeping problems could be present. The nurses described that their minds and thoughts were constantly busy processing impressions from the major incident, as a kind of background music, making it hard to relax. Additionally, cognitive problems, such as having difficulties concentrating, were reported. Most mental reactions started after some time and were more intense a few days after the incident.

> I was so tired. Extremely tired. Had some pain in my body. Needed to rest. And then…the thought went like in a loop- coming back all the time; what you saw, what you did. I kind of went through the situation all over again and again. It lasted for some day maybe. Then I could move forward and put it behind me.

The experience of being exhausted was accompanied by feelings of being vulnerable. Being faced with how the incident had caused personal tragedies and affected human people reminded them of the fragility of life and made the nurses feel vulnerable.

> But then…. after a while. It was very hard actually. But it took three days until I could take it in: those things I've seen there in the bus. I read about the victims in a newspaper. Their families spoke about them, of course. Then I felt….it was tough…I became sad.

The nurses sometimes put themselves in the position of the injured or family members, as if the incident had affected themselves or their own lives. That increased feelings of being sad and powerless. Sometimes, existential thoughts occurred, such as questioning the meaning of life or the meaning of the incident that had occurred.

### Getting distance from the situation

After having overcome the exhaustion, the nurses strived to get distance from the major incident. For some, this phase started only a few hours after the incident, whereas for others, it could be some days after the incident. Being alone, taking one's time to reflect on the actual incident as well as one's own emotions and thoughts, and finding one's own 'inner place' were essential and contributed to recovery. Another strategy used was to return as soon as possible to everyday life and routines, both personal and professional. Such activities and contexts included everyday practices such as eating together and watching

TV with colleagues, including those they had worked with during the major incident, as a method of 'normalisation'. Using humour, for example, giggling together about other things that had occurred during the incident, gave them distance from the experiences.

> For me, it was taking a shower, getting the ambulance uniform off, eat and walk. We laughed a lot during the dinner. We ordered food, and sat around the table and in fact, it was a kind of rambunctious atmosphere. I think it a kind of art, to be able to laugh and relax, despite what has happened. It releases the stress….

Several of the nurses described activities that could be understood as rituals to handle the transition from being a professional during the major incident to becoming a private person. An example of such rituals was taking a bath or shower to cleanse oneself of the experienced situation.

> They had made sandwiches and so, but I could not eat. My body was on high alert so to say, all I wanted was to take a shower, to wash the incident off myself, to clean myself from it in some way…. I did not want to talk about it actually….

Most of the nurses claimed that they preferred to stay with their colleagues and to keep on working after the major incident, even if offered to go home. Staying at work contributed to normalisation and de-dramatised the situation.

### Establishing supportive conditions for recovery

A significant part of the individual responsibility was to create good conditions for their own recovery. However, the preferred conditions varied widely. Participants with less professional experience described how they were unprepared for the time and energy required to recover. Creating their 'own story' of the incident and their efforts sometimes required information from other perspectives or persons involved. One example of such information needed could be information about the victims' outcome.

> Well, for me, I had the whole picture, from alarm to the end so to say, but many of our colleagues had only arrived and picked up a patient they took to the hospital. And they needed to get information on what had happened before they arrived….and I needed information on what had happened to the patient during the transport. So, we spend some time to talk about these things, to clear the scenario out for everyone

If there were fatalities, some nurses conducted private rituals, such as lighting a candle for the victims. Such rituals could serve as an endpoint of the major incident. Creating possibilities of informal sharing of experiences with others who understood the context, such as colleagues or first-line managers, was a crucial part of the supportive environment. Such sharing also contributed to a feeling of being part of the team and getting confirmation of one's efforts during the major incident.

> I think that it is very important with the colleagues, especially those who have long experience, who have seen most things and you can say anything—they can bear to hear it. Because they have seen it themselves, some time. I cannot tell anyone else; they will not understand, and they will distance themselves. I know that my colleagues understand when I have to make a joke about something horrible, because there is no other way to cope with it…That is a respect for the human being as I don't recognize from any other profession.

Almost all nurses reflected on individual self-reflection and informal social support compared with organised psychosocial support, such as after-action reviews or group sessions including psychological debriefing. Many of the nurses considered such organised sessions not to be supportive of their recovery, and for some, they had an opposite effect than intended. Additionally, individuals who did not feel like talking or being around colleagues at all had to be respected. A general caring attitude from colleagues and first-line managers was important to create trust that if someone needed professional support, such support would be immediately provided. Because most mental reactions occurred after some time following the incident, many participants meant that the concern and care from colleagues and managers also had to be present after some time and that offers of professional support were more valuable after some time than during the acute phase.

### Personal and professional development

The experiences gained both from the major incident itself but also from reactions and the following recovery process needed to be integrated into the personal and professional identity and could then contribute to professional and sometimes personal growth. Some nurses stated that this process of incorporating the major event as a part of their overall experiences was a skill that could be learnt over years within the emergency medical services. To facilitate this transformation, continuing to take part in ordinary work, accepting the distress and relying on the individual recovery process were supportive. Senior colleagues and managers could play an important role in this process by confirming the occurrence of distress and sharing their experiences of similar recovery processes. At the same time as it was perceived as important to finalise and 'store' their experiences, the nurses expressed worries about what impact such strong experiences would have on them as both professionals and people in a longer perspective.

> My fiancé came home and just held me. We didn't talk, it wasn't necessary. He just held me. I thought that I had to quit this job. But at the same time; it is a part of me, who I am. I want to be in situations

like this. I want to help. I'm an ambulance nurse, in my person. And so is he. It's part of our lives, that we understand each other's needs in situations like this. I was pregnant that day. He was the only person who knew. I was so worried for the future….

Over time, new perspectives on life, on being a human and on the role of the emergency medical services developed, mostly adding wisdom and being more resilient, as both a person and as a nurse. Most often, this process was positive, leading to an increased self-awareness and ability to manage and endure severe situations.

Some new people that come to us think that it is a tough environment within the ambulance services, that we are hard as stone. I say it is the opposite. We witness a lot, we meet people in all kinds of situations, when they are at most vulnerable, when they are between life or death, when they don't want to live any more, when they are severely beaten…and that could happen to us all. Everyone. Life is not always clean and tidy, not always fair. You realize that after some years. We can laugh about it, together, about things that are not really fun for anyone else. Undramatized it. We live today, but you never know about tomorrow… You learn that within the ambulance services. I think it is good. You need that reminder sometimes. And I live a richer life thanks to that.

However, sometimes, the burden of the experiences from the major event was too hard to endure, and the recovery process did not end with a positive development. In such situations, the nurses felt overwhelmed and unsuccessful, which led to persistent physical or mental health problems and in some cases, an ending of their career.

My colleague, she was incident commander in an accident where two children died. She left the ambulance services after that. She just cried as soon as she came to the work. It was very hard to see her, actually. She was a very competent ambulance nurse, but this crushed her. I tried, we tried all of us. But she took it very personally, she withered away…

The risk for such outcomes was experienced as higher if the nurse was unexperienced and lacked proper formal or mental preparedness for being faced with the complexity and demands of a major incident.

## DISCUSSION

Being deployed as an ambulance nurse in a major incident led physical and psychological health problems and feelings of being exhausted and vulnerable. To recover, the ambulance nurses took responsibility and strived to find strategies to gain distance from the situation and to create supportive conditions. If successful, the recovery process led to professional and personal growth. However, being deployed in major incidents without significant preparedness or experience could harm individuals and, in the worst case, end their career.

In this study, no serious physical injuries were reported, but despite that, many participants described physical health problems and exhaustion. Although physical well-being has been found to be an important mediator for psychological well-being,[8] the physical status of ambulance nurses is important when discussing recovery after working in stressful situations.

The nurses demonstrated a high level of awareness of the occurrence of stress and the risks for being traumatised but also self-knowledge and strategies to manage distress. At the same time, several expressed worries about colleagues and sometimes fear of being exposed to situations that might overload their capacity to recover. A realistic appraisal and active decision-making on how to cope with the situation can contribute to resilience after stressful events.[7] In this study, the participants described different activities that can be seen as 'rituals', striving for balance concerning professional boundaries between themselves as a person and being an ambulance personnel. It seems that these 'rituals' also mediated a transition from 'being in the major incident' to 'being after the incident' and moving to a more ordinary everyday life. In the theoretical framework of transition,[14] this process has been described as a passage from one phase, condition or status to another or as a period between relative stable phases. The transition process is characterised by disconnecting with previous contexts and believes, moving through a period of unfamiliarity and finally the replacement of an existing set of expectations with new ones.[15] The transition, as well as resilience, is depended not only on the individual, but on the social contexts. For a successful transition process, the person needs to have awareness of both their situation and the transition process, and the level of engagement is essential. To develop skills for transition, both supportive measures such as mentorship has been suggested, but also, being exposed to regular transitioning prepare individuals to cope with challenges.[15] Future studies are suggested to explore the relationship between transition and resilience among health professionals exposed to potentially traumatic events.

The participants in this study expressed a strong commitment and convention that they were accountable themselves for creating their own recovery process. Being aware of one's own responsibility for one's well-being and using potential traumatic events and severe stress as a foundation to learn and grow from works of Southwick and Charney[8] can be applied to the results of this study. All transitions or changes also require time,[13] which was strongly emphasised by the study participants who expressed that they sometimes were surprised that their own recovery took longer than expected.

Self-care has been found to be a key to protect against negative health impacts.[16] The study also confirmed that ambulance nurses need to acquire a unique set of personal qualities to promote self-care and self-resilience in their everyday work.[17] A question

raised was whether the competence to develop self-care strategies and use experiences to grow is something that can be developed over time or whether it is a personal attribute needed to be comfortable in a profession such as being a nurse within the ambulance services. Is there a self-selection component that can support resilience within the ambulance services, or is it a result of a professional development process over time?

To mitigate the negative health effects and promote physical and mental well-being requires an awareness of the demands as well as the recovery process. Prevention of work-related traumatic stress should rely on a sound psychosocial work environment, systematic training of employees, social support from colleagues and managers and proper follow-up of employees after a critical event.[18] Well-functioning supportive structures and working environment have been suggested to compensate for short professional experiences[16] and might therefore be of extra importance for less experienced ambulance personnel. Social support seems particularly powerful to reduce negative outcomes of stress.[16 19 20] In line with evidence, organised sessions with the aim to process emotions in the immediate aftermath of the potentially traumatic event, such as debriefings or group counselling, were not experiences that contributed to recovery or resilience. In accordance with previous evidence, such interventions should therefore not be part of the organisational support systems after being deployed to major incidents.[21 22] Instead, this study emphasises the employer's responsibility to understand the recovery process, to enable time and space for the recovery, and to systematically and with empathy follow-up on both physical and mental health, both in the acute phase and over time. Ensuring that ambulance personnel are 'fit for duty', in terms of being physically and mentally ready to respond to challenging situations and also being prepared to recover after such situations, after being deployed in major incidents should therefore be a shared responsibility between the individual professional and the employer as a natural component in a resilient ambulance service both in the every day services and in major incidents. Also, further scientific interest in what demands that are required to respond to such incidents and how such qualities develop is needed.

## Limitations

As in all qualitative research studies, it can never be excluded that another study sample, the characteristics of the major events forming the basis of this study, or the timing of the data collection might have influenced the results. However, the study sample size was considered to have information power,[23] influenced by recruiting study participants with quite recent experiences, high-quality dialogue during the interviews, the two-session interviews and the use of a systematic analysis methodology. None of the included nurses reported worrisome levels of traumatic stress according to the screening. However, it

may be that nurses who have left the ambulance services following a potentially traumatic experiences were less likely to be included in this study, given the recruitment procedure, and this must be considered as a limitation.

The two-session interview design was chosen to ensure that all data were collected and to ensure the well-being of the study participants. The second interviews were in general shorter than the first ones, but still contributed to rich data by confirming the experiences that were expressed in the first interviews and to verifying the researchers' interpretations of the first interview statements. No new topics were expressed in the second interviews. All data were analysed and confirmed by researchers with different perspectives on the topic, including an extensive preunderstanding of the ambulance services, major incidents and caring in emotionally challenging situations, which was found to contribute to the understanding of the data.

## CONCLUSION

The ambulance nurses demonstrated a high level of awareness of the occurrence of stress and individual responsibility to find strategies to recover. A successful recovery from the physical and mental exhaustion experienced after being deployed in a major incident required both individual abilities and self-care strategies as well as a supportive working environment. Organised sessions such as debriefings or group counselling did not contribute to recovery and should therefore not be part of the routine organisational support systems. Instead, supporting individual recovery strategies and following up on physical and mental well-being both in the acute phase and over time should be part of all ambulance services procedures after major incidents to promote individual and organisational resilience. Since the resilience of ambulance personnel is essential for a well-functioning emergency services, further research on how to promote such resilience is strongly needed.

**Acknowledgements** The authors would like to thank Dr Samuel Edelbring for his valuable contribution to the initial design and planning of the data collection for this study.

**Contributors** KB: conceptualisation, analysis, writing of the manuscript, KH: conceptualisation, data collection, writing of the manuscript. Both authors read and approved the final manuscript. KH is the guarantor of the paper.

**Funding** The authors have not declared a specific grant for this research from any funding agency in the public, commercial or not-for-profit sectors.

**Competing interests** None declared.

**Patient and public involvement** Patients and/or the public were not involved in the design, or conduct, or reporting or dissemination plans of this research.

**Patient consent for publication** Not required.

**Ethics approval** This study involves human participants. The study was reviewed and approved by the Swedish Ethical Review Authority (ref id 2018:232).

**Provenance and peer review** Not commissioned; externally peer reviewed.

**Data availability statement** No data are available. The data analysed during the current study are not publicly available due to Swedish laws on research ethics and the ethical approval provided by the Swedish Ethical Review Authority.

**ORCID iD**
Karin Hugelius http://orcid.org/0000-0003-0534-4593

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
