## [Reviewer comments · BMJ Open]

ARTICLE DETAILS

TITLE (PROVISIONAL)	Health and wellbeing after being deployed in a major incident; how do Swedish ambulance nurses perceive their health recover process? A qualitative study.
AUTHORS	Blomberg, Karin; Hugelius, Karin

VERSION 1 – REVIEW

REVIEWER	Gyllencreutz, Lina Department of Nursing, Umeå University
REVIEW RETURNED	03-Mar-2023

GENERAL COMMENTS	Thank you for the opportunity to review the manuscript “Perceived health and the recovery process after being deployed as an ambulance nurse in a major incident – a qualitative study”. It is an interesting manuscript that adds valuable knowledge within the field of health and resilience in ambulance care. The topic is highly important especially based on the high burden on the health care system. I suggest some minor clarifications to improve your work. Methods. Please clarify if you define paramedic as nurses or specialized nurses or if you excluded them in the study sample (page 4, line 37-40 vs line 44-45). Please clarify if this is a national wide study or if the invitations via social media etc. only was made in a specific region. It is not clear to me. Due to the participant recruiting. How many nurses showed interest and were contacted by a researcher? Please also consider and discuss in the methodological limitations that your sample do not include those nurses that have authentic experiences from being deployed in a major incident but are not yet in the organization due to eg. sick leave or that they have ended their employment. Notable is also that none of your participants had remaining traumatic stress which might need to be discussed as well. What about those that have? Concerning your data collection method. It would be interesting to know what the second interview contributed with. Please add information of what the participants added on about the dialogue and well-being in the follow up interview? Results. The first and the second quote at page 8 are in my opinion very similar. You might choose a quote that describes the rituals in a more varied way. This might be even more important as you don't give information about the origin of the quotes (see ethical considerations). You might add information preferable in the method, how many interviews that the quotes represent.
---

	Discussion. The theoretical framework of transition seems relevant but are not enough discussed. Please deepen the part of the framework in the discussion and also relate it to resilience. Conclusion. I believe that this result could be transferable to other incidents than major ones and thus the conclusion should call for more research within the area not exclusive for major incidents but also non-major incidents that are considered particularly challenging eg. calls that consider severe injured children. Thank you very much for a study that add valuable knowledge within an important area.
--	--

REVIEWER	Phung, Viet-Hai University of Lincoln, Community and Health Research Unit, School of Health and Social Care
REVIEW RETURNED	20-Mar-2023

GENERAL COMMENTS	This is a worthy study that discusses an important area in prehospital research, the well-being of staff delivering prehospital care. There are areas within the study that need further elaboration, in relation to the methodology and some definitions (see attached file in the annotations). It would enhance the paper if it drew on some of the findings on the Phung (2022) study on sickness and wellbeing among ambulance personnel.
--

REVIEWER	Wilson, Caitlin University of Leeds
REVIEW RETURNED	29-Mar-2023

GENERAL COMMENTS	Dear authors, Thank you for allowing me to submit your qualitative study manuscript on 'perceived health and the recovery process after being deployed as an ambulance nurse in a major incident'. I have enjoyed reviewing it and believe it to be a well-written manuscript on an interesting topic that has been under-researched. I am recommending revisions and have included below some comments that may help you in revising the manuscript. I am a clinical academic paramedic and I'm offering these comments from a position of wanting to further improve the manuscript. I completely respect that you may wish to disagree with any of the comments I have made, which would be absolutely fine as these are merely my own personal thoughts.  • Abstract setting: I would argue that the setting is the ambulance service in Sweden while 'prehospital major incidents' is the objective/topic so I wouldn't include that here • Abstract results: "Being deployed in major incidents was felt/believed/perceived to be straining..." please add either felt, believed, perceived or similar to clearly get across the qualitative nature of this study • Methods - study sample and recruiting: I would replace 'authentic experiences' with 'first-hand experience' • Methods - study sample and recruiting: I wonder if the authors could include whether they defined a 'major incident' in their recruitment/advertising materials? And if yes, then include this definition here.
---

	 • Methods – analysis: I would like to see a statement on researcher positionality • Patient and Public involvement: I would rephrase to 'no patients were involved in this study' or 'no patient involvement' • I wonder if the authors could reflect on the timing of data collection (i.e. before and after the start of the COVID19 pandemic) and whether any participants mentioned this as a 'major incident'? And subsequently whether they believe this influenced data collection/analysis? • Table 2: Could this maybe be reformatted into a more visually appealing figure rather than a table? I feel like a table isn't particularly necessary but this may be due to personal preference for visual information. • What are the advantages of a two-session interview design versus the usual single interviews? I'm intrigued by this choice of methods and would embrace some further justification/reflection on this aspect. I can see you have briefly reflected on this in the limitations but wonder if it would be better placed in the methods section. • Could you please insert the participant ID numbers after each quote? This would assist the reader in seeing whether there's a good spread of quotes from different participants and build up a better picture of individual and collective views. It would also enable readers to look back at Table 1 for individual participant characteristics if they wanted more background information to a quote such as the type of incident in focus for the interview or years of experience or profession or age. • "None of the participants screened indicated remaining traumatic stress according to the PCL-IV" – could you reflect on whether this was expected and compare this to existing literature • End of discussion section: "In line with evidence, organized sessions such as debriefings or group counselling were not experiences that contributed to recovery or resilience and should therefore not be part of the organizational support systems" – My initial reaction was that I wasn't entirely comfortable with the recommendation that these should not be part of organisational support systems based on your qualitative findings. Upon looking up reference 19 (Cochrane review) and 20 (international guideline) I think this recommendation can be left in but wondered if the authors could add a few more sentences relating to the two references to support that their recommendation is not built just on their qualitative findings but actually is supported in a systematic review and international guideline. • Would the authors consider adding another sentence after 'fit for duty' summarizing what that might look like or highlighting this as an area for future research?
--	---

VERSION 1 – AUTHOR RESPONSE

Reviewer: 1

Dr. Lina Gyllencreutz, Department of Nursing, Umeå University Comments to the Author:

Thank you for the opportunity to review the manuscript "Perceived health and the recovery process after being deployed as an ambulance nurse in a major incident – a qualitative study". It is an interesting manuscript that adds valuable knowledge within the field of health and resilience in ambulance care. The topic is highly important especially based on the high burden on the health care system. I suggest some minor clarifications to improve your work.

Reply: Thank you for this encouraging statement and for taking your time to review our paper.

Methods.

Please clarify if you define paramedic as nurses or specialized nurses or if you excluded them in the study sample (page 4, line 37-40 vs line 44-45).

Reply: Thank you for notifying us on this. We have only included nurses (specialized or not) in the study sample. This has been clarified in the inclusion- and exclusion criteria.

Please clarify if this is a national wide study or if the invitations via social media etc. only was made in a specific region. It is not clear to me.

Reply: Thank you for observing this. Yes, the invitations for the study posted in social media were nationally distributed. This has now been added in the text.

Due to the participant recruiting. How many nurses showed interest and were contacted by a researcher? Please also consider and discuss in the methodological limitations that your sample do not include those nurses that have authentic experiences from being deployed in a major incident but are not yet in the organization due to eg. sick leave or that they have ended their employment.

Reply: Thank you for observing this. All who showed interest to participate were included, and this has been added to the result section. The inclusion of nurses who had ended their employment or was on temporary leave is interesting. However, these could still get the invitations through social media, and we had actually one who was on maternal leave at the time for the data collection. However, we agree that it is a limitation and most likely, nurses who have left the ambulance services due to potentially traumatic events are less likely to show interest for a study like this. We have added this in the section of limitations.

Notable is also that none of your participants had remaining traumatic stress which might need to be discussed as well. What about those that have?

Reply: Thank you for addressing this. We have added this as a limitation in the limitation section.

Concerning your data collection method. It would be interesting to know what the second interview contributed with. Please add information of what the participants added on about the dialogue and well-being in the follow up interview?

Reply: Thank you for addressing this. We have added more information on the two-session method in the limitation section.

Results.

The first and the second quote at page 8 are in my opinion very similar. You might choose a quote that describes the rituals in a more varied way. This might be even more important as you don't give information about the origin of the quotes (see ethical considerations). You might add information preferable in the method, how many interviews that the quotes represent.

Reply: Thank you for good suggestions this. We have changed one of the quotations and added information on how many individuals that has been quoted in the paper.

Discussion.

The theoretical framework of transition seems relevant but are not enough discussed. Please deepen the part of the framework in the discussion and also relate it to resilience.

Reply: We also find the transition framework interesting in terms of resilience. We have discussed the topic deeper and added a reference to this discussion, and also added a thought about future research on how these two phenomena are related.

Conclusion.

I believe that this result could be transferable to other incidents than major ones and thus the conclusion should call for more research within the area not exclusive for major incidents but also non-major incidents that are considered particularly challenging eg. calls that consider severe injured children.

Thank you very much for a study that add valuable knowledge within an important area.

Reply: Thank you for this statement. We agree, and have revised the conclusion in line with your comments.

Reviewer: 2

Dr. Viet-Hai Phung, University of Lincoln Comments to the Author:

This is a worthy study that discusses an important area in prehospital research, the well-being of staff delivering prehospital care. There are areas within the study that need further elaboration, in relation to the methodology and some definitions (see attached file in the annotations). It would enhance the paper if it drew on some of the findings on the Phung (2022) study on sickness and wellbeing among ambulance personnel.

Reply: Thank you for encouraging feedback, and congratulations to your work on health and wellbeing on UK ambulance personnel. Since our paper specifically focus on major incidents, we have included other publications that report on such contexts.

Reviewer: 3

Dr. Caitlin Wilson, University of Leeds

Comments to the Author:

Dear authors,

Thank you for allowing me to submit your qualitative study manuscript on 'perceived health and the recovery process after being deployed as an ambulance nurse in a major incident'. I have enjoyed reviewing it and believe it to be a well-written manuscript on an interesting topic that has been under-researched. I am recommending revisions and have included below some comments that may help you in revising the manuscript. I am a clinical academic paramedic and I'm offering these comments from a position of wanting to further improve the manuscript. I completely respect that you may wish to disagree with any of the comments I have made, which would be absolutely fine as these are merely my own personal thoughts.

Reply: Thank you for taking your time and efforts to improve our paper, it is very appreciated!

- Abstract setting: I would argue that the setting is the ambulance service in Sweden while 'prehospital major incidents' is the objective/topic so I wouldn't include that here
- Abstract results: "Being deployed in major incidents was felt/believed/perceived to be straining..." please add either felt, believed, perceived or similar to clearly get across the qualitative nature of this study

Reply: Thank you for these suggestions that helped us to clarify the study. We agree and have revised in accordance.

- Methods - study sample and recruiting: I would replace 'authentic experiences' with 'first-hand experience'

Reply: We agree with your suggestion and have revised in accordance.

- Methods - study sample and recruiting: I wonder if the authors could include whether they defined a 'major incident' in their recruitment/advertising materials? And if yes, then include this definition here.

Reply: Thank you for comment on this question. We used the same definition in the study recruitment material, as stated under the heading of study setting in the paper.

- Methods – analysis: I would like to see a statement on researcher positionality

Reply: Thank you, however, we are not fully sure what you mean with researcher positionality. In the limitation section, we discuss our preunderstanding and relation to the topic, if that is what is requested. Please, this comment needs to be clarified.

- Patient and Public involvement: I would rephrase to 'no patients were involved in this study' or 'no patient involvement'

Reply: Thank you for this supportive suggestion. We agree with your suggestion and have revised in accordance.

- I wonder if the authors could reflect on the timing of data collection (i.e. before and after the start of the COVID19 pandemic) and whether any participants mentioned this as a 'major incident'? And subsequently whether they believe this influenced data collection/analysis?

Reply: Thank you for this comment. Data were collected during the period of December 2018 to February 2020, and therefore, there were no "covid- 19 effects" on this study.

- Table 2: Could this maybe be reformatted into a more visually appealing figure rather than a table? I feel like a table isn't particularly necessary but this may be due to personal preference for visual information.

Reply: We appreciate your comment! We think that both the table and the figure illustrates the results and contributes to an understanding of the results.

- What are the advantages of a two-session interview design versus the usual single interviews? I'm intrigued by this choice of methods and would embrace some further justification/reflection on this aspect. I can see you have briefly reflected on this in the limitations but wonder if it would be better placed in the methods section.

Reply: Thank you for your suggestions. In accordance with suggestions from one of the other reviewer, we have added a deeper reflection on the two-session interview technique in the limitation section.

- Could you please insert the participant ID numbers after each quote? This would assist the reader in seeing whether there's a good spread of quotes from different participants and build up a better

picture of individual and collective views. It would also enable readers to look back at Table 1 for individual participant characteristics if they wanted more background information to a quote such as the type of incident in focus for the interview or years of experience or profession or age.

Reply: Thank you for your suggestion. In accordance with suggestions from the editor and other reviewers, the Table 1 has been removed and in order to protect the identity of our study participants, we have actually removed more information on them in the paper. However, we have added information that the quotations replies to 7 individuals, but to keep their identities, we stay with the decision not to specify what quotation that is related to each study participant.

- “None of the participants screened indicated remaining traumatic stress according to the PCL-IV” – could you reflect on whether this was expected and compare this to existing literature

Reply: Thank you for noticing this. We think it was expected, given the scenarios that the nurses had been exposed to, but the screening was made as both a result but also as part of the concern for the study participants. Some further reflections on the screening results and potential biases in the study sample has been added in the discussion.

- End of discussion section: “In line with evidence, organized sessions such as debriefings or group counselling were not experiences that contributed to recovery or resilience and should therefore not be part of the organizational support systems” – My initial reaction was that I wasn’t entirely comfortable with the recommendation that these should not be part of organisational support systems based on your qualitative findings. Upon looking up reference 19 (Cochrane review) and 20 (international guideline) I think this recommendation can be left in but wondered if the authors could add a few more sentences relating to the two references to support that their recommendation is not built just on their qualitative findings but actually is supported in a systematic review and international guideline.

Reply: Thank you for your suggestion. We have revised the section and hope that it is clearer and better expressed now.

- Would the authors consider adding another sentence after ‘fit for duty’ summarizing what that might look like or highlighting this as an area for future research?

Reply: Thank you for your suggestion. We have revised the section and hope that it is clearer and better expressed now.

VERSION 2 – REVIEW

REVIEWER	Phung, Viet-Hai University of Lincoln, Community and Health Research Unit, School of Health and Social Care
REVIEW RETURNED	06-Apr-2023

GENERAL COMMENTS	Generally well-written in an important area of work. Just a few minor suggested revisions/clarifications needed. Not sure the abstract needs a summary. The abstract should itself accurately summarise the content of the article. Abstract - Results - "perceived to be" not "perceived as".
--

	Study sample and recruiting - replace "national wise" with "nationwide". Explain why the cut off date for recruitment was five years (where a major incident had occurred). Data collection - elaborate on why there was such an imbalance between the number of face-to-face and telephone interviews. Analysis - explain why you preferred thematic analysis to framework analysis. Results - why did you include an interview lasting only two minutes?
--	--

REVIEWER	Wilson, Caitlin University of Leeds
REVIEW RETURNED	06-Apr-2023

GENERAL COMMENTS	Thank you for taking the time to respond to my earlier comments and make changes to the manuscript as you saw fit. I would now recommend this manuscript for acceptance.
--

VERSION 2 – AUTHOR RESPONSE

Reviewer: 2

Dr. Viet-Hai Phung, University of Lincoln Comments to the Author:

Generally well-written in an important area of work.

Just a few minor suggested revisions/clarifications needed.

Not sure the abstract needs a summary. The abstract should itself accurately summarise the content of the article.

Abstract - Results - "perceived to be" not "perceived as".

Reply: Thank you for notifying us on this, we have changes in accordance with your suggestion.

Study sample and recruiting - replace "national wise" with "nationwide". Explain why the cut off date for recruitment was five years (where a major incident had occurred).

Reply: Thank you for this comment. Since we have got a notice from the editor to limit the paper to we have reduced many sections and therefore, after considerations we need to stay with the current description of the inclusion criteria.

Data collection - elaborate on why there was such an imbalance between the number of face-to-face and telephone interviews.

Reply: Thank you for this comment. We have changes in accordance with your suggestion.

Analysis - explain why you preferred thematic analysis to framework analysis.

Reply: Thank you for asking this. The thematic analysis was chosen with regard to the data characteristics and since it is a well established method to use. Since we have not get this requirement from any other reviewer and we have to reduce the number of words, we have chosen not to further elaborate on this question.

Results - why did you include an interview lasting only two minutes?

Reply: Thank you for this notification. Since we used a two step datacollection with almost all study persons (as described), the second interview was used both to confirm data from the first interview and to ensure the well-being of the study participants. The second interviews were in shorter than the first ones, and even if two minutes seems very short, we find it necessary to include all interviews and not exclude anything from the data collection. We also think it is transparent to present the length of all interviews, even the very short ones.

Reviewer: 3

Dr. Caitlin Wilson, University of Leeds

Comments to the Author:

Thank you for taking the time to respond to my earlier comments and make changes to the manuscript as you saw fit. I would now recommend this manuscript for acceptance.

Reply: Thank you very much for your time and efforts!

VERSION 3 – REVIEW

REVIEWER	Phung, Viet-Hai University of Lincoln, Community and Health Research Unit, School of Health and Social Care
REVIEW RETURNED	12-May-2023
GENERAL COMMENTS	The reviewers' responses to my original comments were rationalised well. I recommend the paper for publication.